# Modeling of High-Power Tonpilz Terfenol-D Transducer Using Complex Material Parameters

**DOI:** 10.3390/s22103781

**Published:** 2022-05-16

**Authors:** Yanfei Wei, Xin Yang, Yukai Chen, Zhihe Zhang, Haobin Zheng

**Affiliations:** School of Electrical and Information Engineering, Hunan University, Changsha 410082, China; wyf_8866@163.com (Y.W.); chenyukai@hnu.edu.cn (Y.C.); zhangzhihe@hnu.edu.cn (Z.Z.); zhenghaobin@hnu.edu.cn (H.Z.)

**Keywords:** Terfenol-D, losses, complex parameters, plane-wave method (PWM), finite element method (FEM)

## Abstract

The loss effect in smart materials, the active part of a transducer, is of significant importance to acoustic transducer designers, as it directly affects the important characteristics of the transducer, such as the impedance spectra, frequency response, and the amount of heat generated. It is therefore beneficial to be able to incorporate energy losses in the design phase. For high-power low-frequency transducers requiring more smart materials, losses become even more appreciable. In this paper, similar to piezoelectric materials, three losses in Terfenol-D are considered by introducing complex quantities, representing the elastic loss, piezomagnetic loss, and magnetic loss. The frequency-dependent eddy current loss is also considered and incorporated into the complex permeability of giant magnetostrictive materials. These complex material parameters are then successfully applied to improve the popular plane-wave method (PWM) circuit model and finite element method (FEM) model. To verify the accuracy and effectiveness of the proposed methods, a high-power Tonpilz Terfenol-D transducer with a resonance frequency of around 1 kHz and a maximum transmitting current response (TCR) of 187 dB/1A/μPa is manufactured and tested. The good agreement between the simulation and experimental results validates the improved PWM circuit model and FEA model, which may shed light on the more predictable design of high-power giant magnetostrictive transducers in the future.

## 1. Introduction

Terfenol-D has a larger magnetostrictive coefficient, higher magneto-mechanical coupling coefficient, and higher energy density than traditional magnetostrictive materials, such as nickel and piezoelectric materials, and due to the use of state-of-the-art SiC devices [1], Terfenol-D has permitted the building of various actuating devices, including actuators, transducers, and motors, both at the macro- and micro-scale. High dynamic strains have been produced in Terfenol-D linear actuators using the device mechanical resonance, even against a high load, where very large powers and good efficiency can be achieved [2]. Due to these excellent properties, a few high-power magnetostrictive underwater transducers already outperform PZT transducers, especially in the low-frequency domain. The design of a giant magnetostrictive transducer has been attracting great interest in the field of underwater acoustic research [3,4].

The design process of acoustic transducers for adequate performance and reliability has increasingly come to rely upon robust modeling techniques and numerical tools of analysis [5]. The existing popular modeling methods of acoustic transducers mainly include the lumped-parameter method, plane-wave method (PWM) circuit model, and finite element method (FEM) model. The lumped-parameter method [6] adopts a single-degree-of-freedom, spring-mass-damping system to represent a Tonpilz transducer, which can easily calculate the resonant frequency of the transducer. However, its accuracy is limited, and it can only be used to roughly estimate the size of the transducer. Based on the lumped-parameter method, Sherman and Butler [7] used electrical components to represent the mechanical system, successfully established a lumped-parameter circuit model of a transducer, and simulated the electrical and mechanical characteristics. According to the circuit theory of wave propagation, such as transmission lines, Tilmans [8] established the PWM model. The accuracy of this model is much higher than that of the lumped-parameter circuit model, and even better accuracy can be achieved under high-frequency conditions. Recently, a PWM equivalent circuit model was successfully used in a longitudinal–torsional ultrasonic transducer [9], a Tonpilz transducer head mass selection according to excitation signal type [10], and analysis of the transmitting characteristics of an acoustic conformal array of multimode Tonpilz transducers [11]. Giant magnetostrictive materials follow the piezomagnetic laws, which are very similar to the piezoelectric laws, in a quasi-linear manner. Therefore, Ackerman [12] deduced the PWM model for Terfenol-D devices. Butler [13] established a PWM equivalent model of a 2.5 kHz high-power Terfenol-D transducer and verified its feasibility.

FEM is widely used as a design tool for transducer design, as it is very capable of dealing with highly complex geometries and calculating the modes of vibration and the coupling between these modes [11]. FEM can also allow accurate calculations of the stress and strain distributions in the structure and calculations of the vibration displacement response at the output surface for a known electrical excitation [14], presenting various results. Recently, efforts were made to model the nonlinearity of material behavior using the curve fitting technique [5] and the discrete energy-averaged model [15]. However, for those modeling techniques, the contribution to the power dissipation of the material constants is usually ignored. However, the loss effect in smart materials, the active part of a transducer, is of significant importance to acoustic transducer designers, as it directly affects the important characteristics of the transducer, such as the impedance spectra, frequency response, and the amount of heat generated [16]. Therefore, only the mechanical behavior of the transducer design can be accurately estimated using those methods. Neglecting the loss effect would cause substantial simulation errors [17]. In order to consider the loss of piezoelectric transducers, Sherrit [18] used a PWM model with complex material constants to represent the piezoelectric material parameters in order to characterize the piezoelectric losses. Recently, a PWM circuit model with dielectric, elastic, and piezoelectric losses was developed by Dong [19,20] to verify the accuracy improvement. Greenough [21] established the equivalent PWM circuit model of Terfenol-D to extract the loss-related material constants. The characterization of Terfenol-D losses, including mechanical, piezomagnetic, and magnetic losses, using complex material parameters is discussed in [22], but the eddy loss is not discussed. For high-power low-frequency Terfenol-D transducers requiring more smart materials, material losses become even more appreciable. To date, there has been few reports on incorporating Terfenol-D complex material parameters into the PWM or FEM modeling techniques.

In conclusion, methods for Terfenol-D applications and finite element simulation methods for giant magnetostrictive transducers taking into account losses are rarely reported. In this paper, a PWM of a high-power giant magnetostrictive transducer considering the losses of Terfenol-D is established. The three complex parameters of the complex compliance coefficient, the complex piezomagnetic coefficient, and complex permeability are used to realize the characterization of losses. To consider the eddy current effect in giant magnetostrictive materials, the variance of equivalent magnetic permeability with frequency is considered. These complex material parameters are also used in the FEM calculation to realize the consideration of losses. To verify the feasibility and accuracy of the proposed PWM and FEM models, a high-power Tonpilz Terfenol-D transducer with a resonance frequency of around 1 kHz and a maximum transmitting current response (TCR) of 187 dB/1A/μPa is manufactured and tested. The good agreement between the simulation and experimental results successfully validates the improved PWM circuit model and FEA model, which may shed light on the more predictable design of high-power giant magnetostrictive transducers in the future.

## 2. The High-Power Tonpilz Terfenol-D Transducer

The structure of a high-power Tonpilz Terfenol-D transducer is mainly composed of a head mass, a drive section, a tail mass, and a stress rod with a metallic bolt to fasten all the components. A 1 kHz Tonpilz giant magnetostrictive transducer was designed and manufactured. A structural diagram of the transducer is shown in Figure 1. This transducer uses high-strain Terfenol-D as the driving material, and the driving section consists of four Terfenol-D rods with a diameter of 20 mm and a length of 100 mm. A samarium–cobalt permanent magnet with a thickness of 15 mm and a diameter of 20 mm is pasted on both ends of each Terfenol-D rod to provide a bias magnetic field of around 45 kA/m. The stress rod passing through the center of the assembly provides a prestress of 20 MPa for the rods by the disc springs. To reduce the eddy current, the Terfenol-D rod is radially slotted, and the schematic diagram of the Terfenol-D rod is shown in Figure 2. In order to provide AC excitation to the rods, 1300 turns of AC drive wound wire electric solenoid are wound on each rod.

The total length of the energy device is *l* = 247 mm, the thickness of the head mass is *l_h_* = 45 mm, the thickness of the tail mass is *l_t_* = 72 mm, the diameter of the head mass *d_h_* and the diameter of the tail mass *d_t_* are both 160 mm, and the transducer mass is around 90 kg. The structural parameters of the transducer are shown in Table 1.

## 3. Loss Integration into PWM Equivalent Circuit

A cylindrical Terfenol-D in *k*_33_ vibration mode is discussed in this paper, as shown in Figure 3. The coordinate system is established with one end of the rod as the origin of coordinates, and the vibration of the Terfenol-D rod is simplified to simple harmonic motion without considering the time response of the circuit; a conventional PWM equivalent circuit without considering material losses [23] is described in Figure 4.

In Figure 3 and Figure 4, $l_{g}$ is the length of the Terfenol-D rod, I denotes the input current, V is the induced electromotive force generated in the loop, and *F*_1_ and *F*_2_ are the forces in the mechanical terminals. *ε*_1_ and *ε*_2_ are the vibration velocity at *z* = 0 and *z* = *l_g_*, respectively. *L*_0_ is the damped inductance and expressed as [23]
(1)$$L_{0}=\frac{N^{2}A\mu_{33}^{S}}{l}$$
(2)$$\mu_{33}^{S}=\mu_{33}^{T}-\frac{d_{33}{}^{2}}{S_{33}^{H}}$$
where $\mu_{33}^{T}$ represents the relative permeability under constant stress, $S_{33}^{H}$ stands for the elastic compliance under constant magnetic field intensity, and $d_{33}$ denotes the piezomagnetic constant.

The electromechanical conversion factor is described as
(3)$$\varphi =\frac{d_{33}}{j\omega N\mu_{33}^{S}S_{33}^{H}}$$

$Z_{g1}$ and $Z_{g2}$ are impedances and are expressed as
(4)$$Z_{g1}=j\rho_{g}c_{g}S_{g}\tan(\frac{k_{g}l_{g}}{2})$$
(5)$$Z_{g2}=\rho_{g}c_{g}S_{g}/j\sin(k_{g}l_{g})$$

Herein, $\rho_{g}$ is the density, the cross-section area is *S_g_* = π*r_g_*^2^, the wave number *k_g_* is denoted by *k_g_* = ω/*c_g_*, ω stands for the frequency, and the sound velocity is expressed as $c_{g}=\sqrt{\frac{1}{\rho gS_{33}^{H}}}$.

An equivalent circuit of the Tonpilz transducer without considering material losses is shown in Figure 5, where the subscripts *r*, *h*, *g*, and *t* are the radiation load, head mass, Terfenol-D rod, and tail mass, respectively. *Z_h_*_1_ and *Z_h_*_2_ denote the series and parallel impedances of the head mass, and *Z_t_*_1_ and *Z_t_*_2_, represent the series and parallel impedances of the tail mass. The detailed impedance expressions of the above components can be found in Table 2.

*Z_r_* is the radiation impedance, *Z_r_* in air can be treated as zero, and the radiation impedance in water is [7]
(6)$$Z_{r}=\rho cA\left[\left(1-\frac{J_{1}(2kR)}{kR}\right)+j\frac{H_{1}(2kR)}{kR}\right]$$
where *J*_1_ is the Bessel function of the first kind, *A* is the area of the vibrating surface, and *H*_1_ is defined as follows:(7)$$H_{1}\approx \frac{2}{\pi }-J_{0}(kR)+\left(\frac{16}{\pi }-5\right)\frac{\sin(kR)}{kR}+(12-\frac{36}{\pi })\frac{1-\cos(kR)}{{\left(kR\right)}^{2}}$$
where *k* is the wave number, and *R* is the radius of the active surface.

According to the equivalent circuit in Figure 5, the mechanical impedance *Z_m_* of the transducer can be
(8)$$Z_{m}=\frac{Z_{m1}*Z_{m2}}{Z_{m1}+Z_{m2}}+Z_{g2}$$

In Equation (8), $Z_{m1}=\frac{(Z_{r}+Z_{t1})*Z_{t2}}{Z_{r}+Z_{t1}+Z_{t2}}+Z_{t1}+Z_{g1}$ and $Z_{m2}=\frac{Z_{h1}*Z_{h2}}{Z_{h1}+Z_{h2}}+Z_{h1}+Z_{g1}$. 

According to the circuit theory, the circuit diagram shown in Figure 5 can be simplified to the circuit diagram shown in Figure 6.

The input impedance of the transducer *Z_e_* is described as
(9)$$Z_{e}=R_{c}+\frac{j\omega L_{0}*Z_{m}/\varphi^{2}}{j\omega L_{0}+Z_{m}/\varphi^{2}}$$
where *R_c_* is the resistance of the wound wire electric solenoid, and $Z_{0}=j\omega L_{0}$.

MATLAB is used to program and simulate Equation (9), and the calculated result of the transducer’s impedance without material losses is shown in Figure 7, which adopts the classic Terfenol-D material parameters as presented in [13].

Assuming that the radiation resistance in the air is 0, ignoring the Terfenol-D losses, the impedance amplitude at resonance is very large, indicating a high Q situation. That is, the Terfenol-D in the PWM model is similar to the lossless transmission line [24]. There is no loss of energy in the vibration process of the transducer, which is not in line with the actual situation. Therefore, the loss during the operation of the giant magnetostrictive transducer must be considered in the modeling process.

### Losses in PWM Equivalent Circuit

Based on the aforementioned PWM equivalent circuit, three hysteresis losses are firstly considered in the giant magnetostrictive rod, namely, the elastic loss, piezomagnetic loss, and magnetic loss. Complex parameters are commonly used to express the losses [22]:(10)$$\left\{ \begin{array}{l} S_{33}^{H}=S_{33}^{H^{\prime }}+jS_{33}^{H^{\prime \prime }}\\ d_{33}=d_{33}^{\prime }+jd_{33}^{\prime \prime }\\ \mu_{33}^{T}=\mu_{33}^{T^{\prime }}+j\mu_{33}^{T^{\prime \prime }} \end{array}\right.$$

The imaginary part represents the mechanical, piezomagnetic, and magnetic losses of the material.

Substituting Equation (10) into Equations (1) and (3), the inductance *L*_1_ and electromechanical conversion coefficient *φ*_1_ can be re-written as follows:(11)$$L_{1}=L_{1}^{\prime }+jL_{1}^{\prime \prime }$$
(12)$$\varphi_{1}=\varphi_{1}^{\prime }+j\varphi_{1}^{\prime \prime }$$
(13)$$L_{1}^{\prime }=\frac{N^{2}A_{g}}{l_{g}}\frac{\mu_{33}^{T^{\prime }}\left(S_{33}^{{H^{\prime }}^{2}}+S_{33}^{{H^{\prime }}^{2}}\right)+S_{33}^{B^{\prime }}\left(d_{33}^{{}^{\prime }2}-d_{33}^{{}^{\prime \prime }2}+2d_{33}^{\prime }d_{33}^{\prime \prime }\frac{S_{33}^{H^{\prime \prime }}}{S_{33}^{H^{\prime }}}\right)}{S_{33}^{H^{{}^{\prime }2}}+S_{33}^{H^{{}^{\prime \prime }2}}}$$
(14)$$L_{1}^{\prime \prime }=\frac{N^{2}A_{g}}{l_{g}}\frac{\mu_{33}^{T^{\prime \prime }}\left(S_{33}^{H^{{}^{\prime }2}}+S_{33}^{H^{{}^{\prime \prime }2}}\right)+S_{33}^{H^{\prime \prime }}\left(d_{33}^{{}^{\prime \prime }2}-d_{33}^{{}^{\prime }2}+2d_{33}^{\prime }d_{33}^{\prime \prime }\frac{S_{33}^{H^{\prime }}}{S_{33}^{H^{\prime \prime }}}\right)}{S_{33}^{H^{{}^{\prime }2}}+S_{33}^{H^{{}^{\prime \prime }2}}}$$
(15)$$\varphi_{1}{}^{\prime }=\frac{d_{33}^{\prime }(\mu_{33}^{T^{\prime }}S_{33}^{H^{\prime }}-\mu_{33}^{T^{\prime \prime }}S_{33}^{H^{\prime \prime }})+d_{33}^{\prime \prime }(\mu_{33}^{T^{\prime }}S_{33}^{H^{\prime \prime }}+\mu_{33}^{T^{\prime \prime }}S_{33}^{H^{\prime }})}{\omega N[{\left(\mu_{33}^{T^{\prime }}S_{33}^{H^{\prime }}-\mu_{33}^{T^{\prime \prime }}S_{33}^{H^{\prime \prime }}\right)}^{2}+{\left(\mu_{33}^{T^{\prime }}S_{33}^{H^{\prime \prime }}+\mu_{33}^{T^{\prime \prime }}S_{33}^{H^{\prime }}\right)}^{2}]}$$
(16)$$\varphi_{1}{}^{\prime \prime }=\frac{d_{33}^{\prime \prime }(\mu_{33}^{T^{\prime }}S_{33}^{H^{\prime }}-\mu_{33}^{T^{\prime \prime }}S_{33}^{H^{\prime \prime }})-d_{33}^{\prime }(\mu_{33}^{T^{\prime }}S_{33}^{H^{\prime \prime }}+\mu_{33}^{T^{\prime \prime }}S_{33}^{H^{\prime }})}{\omega N[{\left(\mu_{33}^{T^{\prime }}S_{33}^{H^{\prime }}-\mu_{33}^{T^{\prime \prime }}S_{33}^{H^{\prime \prime }}\right)}^{2}+{\left(\mu_{33}^{T^{\prime }}S_{33}^{H^{\prime \prime }}+\mu_{33}^{T^{\prime \prime }}S_{33}^{H^{\prime }}\right)}^{2}]}$$

Under the excitation of an alternating magnetic field, an eddy current exists in the giant magnetostrictive material, and the eddy current loss varies with frequency, which does not exist in the popular piezoelectric materials. Therefore, the magnetic loss of the giant magnetostrictive transducer ought to be frequency dependent. However, the imaginary part of the complex permeability in Equation (10) does not change with frequency, which is insufficient to characterize the frequency-dependent eddy current loss. Therefore, it is necessary to incorporate the eddy current loss factor in order to characterize the dynamic characteristics of the eddy current loss [25]. Equation (10) can be expressed as
(17)$$\left\{ \begin{array}{l} S_{33}^{B}=S_{33}^{B^{\prime }}+jS_{33}^{B^{\prime \prime }}\\ d_{33}=d_{33}^{\prime }+jd_{33}^{\prime \prime }\\ \mu_{33}^{T}=\mu_{33}^{T^{\prime }}(\chi_{r}+j\chi_{i})+j\mu_{33}^{T^{\prime \prime }} \end{array}\right.$$

Substituting Equation (17) into Equations (1) and (3), the inductance *L*_2_ and electromechanical conversion coefficient *φ*_2_ can be obtained as follows:(18)$$L_{2}=L_{2}^{\prime }+jL_{2}^{\prime \prime }$$
(19)$$\varphi_{2}=\varphi_{2}^{\prime }+j\varphi_{2}^{\prime \prime }$$
(20)$$L_{2}^{\prime }=\frac{N^{2}A_{g}}{l_{g}}\frac{\mu_{33}^{T^{\prime }}\chi_{r}\left(S_{33}^{H^{{}^{\prime }2}}+S_{33}^{H^{{}^{\prime \prime }2}}\right)+S_{33}^{B^{\prime }}\left(d_{33}^{{}^{\prime }2}-d_{33}^{{}^{\prime \prime }2}+2d_{33}^{\prime }d_{33}^{\prime \prime }\frac{S_{33}^{H^{\prime \prime }}}{S_{33}^{H^{\prime }}}\right)}{S_{33}^{H^{{}^{\prime }2}}+S_{33}^{H^{{}^{\prime \prime }2}}}$$
(21)$$L_{2}^{\prime \prime }=\frac{N^{2}A_{g}}{l_{g}}\frac{\left(\mu_{33}^{T^{\prime \prime }}-\mu_{33}^{T^{\prime }}\chi_{i}\right)\left(S_{33}^{H^{{}^{\prime }2}}+S_{33}^{H^{{}^{\prime \prime }2}}\right)+S_{33}^{B^{\prime \prime }}\left(d_{33}^{{}^{\prime \prime }2}-d_{33}^{{}^{\prime }2}+2d_{33}^{\prime }d_{33}^{\prime \prime }\frac{S_{33}^{H^{\prime }}}{S_{33}^{H^{\prime \prime }}}\right)}{S_{33}^{H^{{}^{\prime }2}}+S_{33}^{H^{{}^{\prime \prime }2}}}$$
(22)$$\varphi_{2}{}^{\prime }=\frac{d_{33}^{\prime }\left(\mu_{33}^{T^{\prime }}\chi_{r}S_{33}^{H^{\prime }}-\left(\mu_{33}^{T^{\prime \prime }}-\mu_{33}^{T^{\prime }}\chi_{i}\right)S_{33}^{H^{\prime \prime }}\right)+d_{33}^{\prime \prime }\left(\mu_{33}^{T^{\prime }}\chi_{r}S_{33}^{H^{\prime \prime }}+\left(\mu_{33}^{T^{\prime \prime }}-\mu_{33}^{T^{\prime }}\chi_{i}\right)S_{33}^{H^{\prime }}\right)}{\omega N\left[{\left(\mu_{33}^{T^{\prime }}\chi_{r}S_{33}^{H^{\prime }}-\left(\mu_{33}^{T^{\prime \prime }}-\mu_{33}^{T^{\prime }}\chi_{i}\right)S_{33}^{H^{\prime \prime }}\right)}^{2}+{\left(\mu_{33}^{T^{\prime }}\chi_{r}S_{33}^{H^{\prime \prime }}+\left(\mu_{33}^{T^{\prime \prime }}-\mu_{33}^{T^{\prime }}\chi_{i}\right)S_{33}^{H^{\prime }}\right)}^{2}\right]}$$
(23)$$\varphi_{2}{}^{\prime \prime }=\frac{d_{33}^{\prime \prime }\left(\mu_{33}^{T^{\prime }}\chi_{r}S_{33}^{H^{\prime }}-\left(\mu_{33}^{T^{\prime \prime }}-\mu_{33}^{T^{\prime }}\chi_{i}\right)S_{33}^{H^{\prime \prime }}\right)-d_{33}^{\prime }\left(\mu_{33}^{T^{\prime }}\chi_{r}S_{33}^{H^{\prime \prime }}+\left(\mu_{33}^{T^{\prime \prime }}-\mu_{33}^{T^{\prime }}\chi_{i}\right)S_{33}^{H^{\prime }}\right)}{\omega N\left[{\left(\mu_{33}^{T^{\prime }}\chi_{r}S_{33}^{H^{\prime }}-\left(\mu_{33}^{T^{\prime \prime }}-\mu_{33}^{T^{\prime }}\chi_{i}\right)S_{33}^{H^{\prime \prime }}\right)}^{2}+{\left(\mu_{33}^{T^{\prime }}\chi_{r}S_{33}^{H^{\prime \prime }}+\left(\mu_{33}^{T^{\prime \prime }}-\mu_{33}^{T^{\prime }}\chi_{i}\right)S_{33}^{H^{\prime }}\right)}^{2}\right]}$$
where $\chi_{r}$ and $\chi_{i}$ are the eddy current factors, which are related to the cut-off frequency *f_c_*. According to [26], the *f_c_* of the Terfenol-D used is 3 kHz. In this case, $f\ll f_{c}$. According to [27], the eddy current factors are
(24)$$\left\{ \begin{array}{l} \chi_{r}=1-\frac{1}{48}{\left(\frac{f}{f_{c}}\right)}^{2}+\frac{19}{30,720}{\left(\frac{f}{f_{c}}\right)}^{4}+\ldots \\ \chi_{i}=\frac{1}{8}\left(\frac{f}{f_{c}}\right)-\frac{11}{3072}{\left(\frac{f}{f_{c}}\right)}^{3}+\frac{473}{4,343,680}{\left(\frac{f}{f_{c}}\right)}^{5}+\ldots \end{array}\right.$$

After considering all the loss factors, a new equivalent circuit is obtained by separating the real and imaginary parts of the Terfenol-D part, as shown in Figure 8, where the parameters are
(25)$$R_{g1}=\rho_{g}A_{g}\frac{c_{g}^{\prime \prime }\sin(k_{g}^{\prime }l_{g})ch(k_{g}^{\prime \prime }l_{g})+c_{g}^{\prime }\cos(k_{g}^{\prime }l_{g})sh(k_{g}^{\prime \prime }l_{g})}{\sin^{2}(k_{g}^{\prime }l_{g})ch^{2}(k_{g}^{\prime \prime }l_{g})+\cos^{2}(k_{g}^{\prime }l_{g})sh^{2}(k_{g}^{\prime \prime }l_{g})}$$
(26)$$X_{g1}=\rho_{g}A_{g}\frac{c_{g}^{\prime \prime }\cos(k_{g}^{\prime }l_{g})sh(k_{g}^{\prime \prime }l_{g})-c_{g}^{\prime }\sin(k_{g}^{\prime }l_{g})ch(k_{g}^{\prime \prime }l_{g})}{\sin^{2}(k_{g}^{\prime }l_{g})ch^{2}(k_{g}^{\prime \prime }l_{g})+\cos^{2}(k_{g}^{\prime }l_{g})sh^{2}(k_{g}^{\prime \prime }l_{g})}$$
(27)$$R_{g2}=\rho_{g}A_{g}\frac{c_{g}^{\prime }sh(k_{g}^{\prime }l_{g})-c_{g}^{\prime \prime }\sin(k_{g}^{\prime }l_{g})}{2\left[\cos^{2}\left(\frac{k_{g}^{\prime }l_{g}}{2}\right)ch^{2}\left(\frac{k_{g}^{\prime \prime }l_{g}}{2}\right)+\sin^{2}\left(\frac{k_{g}^{\prime }l_{g}}{2}\right)sh^{2}\left(\frac{k_{g}^{\prime \prime }l_{g}}{2}\right)\right]}$$
(28)$$X_{g2}=\rho_{g}A_{g}\frac{c_{g}^{\prime }\sin(k_{g}^{\prime }l_{g})+c_{g}^{\prime \prime }\sin(k_{g}^{\prime }l_{g})}{2\left[\cos^{2}\left(\frac{k_{g}^{\prime }l_{g}}{2}\right)ch^{2}\left(\frac{k_{g}^{\prime \prime }l_{g}}{2}\right)+\sin^{2}\left(\frac{k_{g}^{\prime }l_{g}}{2}\right)sh^{2}\left(\frac{k_{g}^{\prime \prime }l_{g}}{2}\right)\right]}$$
(29)$$k=k^{\prime }+jk^{\prime \prime }=\omega {\left[\frac{\rho S_{33}^{H^{\prime }}}{2}(D+1)\right]}^{\frac{1}{2}}-j\omega {\left[\frac{\rho S_{33}^{H^{\prime \prime }}}{2}(D-1)\right]}^{\frac{1}{2}}$$
(30)$$c=c^{\prime }+jc^{\prime \prime }={\left(\rho S_{33}^{H^{\prime }}\right)}^{-\frac{1}{2}}\frac{1}{D}{\left[\frac{1}{2}(D+1)\right]}^{\frac{1}{2}}+j{\left(\rho S_{33}^{H^{\prime \prime }}\right)}^{-\frac{1}{2}}\frac{1}{D}{\left[\frac{1}{2}(D-1)\right]}^{\frac{1}{2}}$$
(31)$$D={\left[1+{\left(\frac{S_{33}^{H^{\prime \prime }}}{S_{33}^{H^{\prime }}}\right)}^{2}\right]}^{\frac{1}{2}}$$
(32)$$Z_{0}=j\omega L_{0}=j\omega (L_{0}^{\prime }+jL_{0}^{\prime \prime })=R_{0}+jX_{0}$$
(33)$$R_{0}=-\omega L_{0}^{\prime \prime }=-\omega \frac{N^{2}A}{l}\mu_{33}^{S}{}^{\prime \prime }$$
(34)$$X_{0}=\omega L_{0}^{\prime }=\omega \frac{N^{2}A}{l}\mu_{33}^{S}{}^{\prime }$$

The impedance of the mechanical part is changed from pure reactance to resistance and reactance, the real part of the inductance of the magnetic circuit part is the inductance representing the energy storage, and the imaginary part is the resistance representing the magnetic loss. The imaginary part of the permeability $\mu_{33}^{S^{\prime \prime }}$ is a negative value. The imaginary part of the inductance $L_{0}^{\prime \prime }$ calculated according to Equations (14) and (21) is also negative. Therefore, the resistance $R_{0}$ obtained by converting the imaginary part of the inductance, representing the magnetic loss, is a positive resistance that varies with frequency. The imaginary part of the material characteristic parameters is eventually converted into the resistances of the PWM equivalent circuit, and, thus, it is expressed as the energy dissipation elements in the equivalent circuit of the giant magnetostrictive transducer. Therefore, the loss can be directly considered if the six characteristic parameters ($S_{33}^{H^{\prime }},S_{33}^{H^{\prime \prime }},d_{33}^{\prime },d_{33}^{\prime \prime },\mu_{33}^{T^{\prime }},\mu_{33}^{T^{\prime \prime }}$) of the giant magnetostrictive transducer material under specific working conditions are determined.

## 4. Loss Integration into FEM

FEM is a numerical technique that can analyze the continuous vibration and deformation of transducers. If accurate material properties are known and are set in the simulation, FEM will have a high prediction accuracy, so FEM is currently the most used analysis method for transducer design and simulation verification. However, due to the limited availability of FEM programs with magnetostrictive elements, the piezoelectric magnetic analogy method is mostly used at present. Reference [7] compares the analogous method of piezoelectric transducers and magnetostrictive transducers in detail, but losses are not considered in the simulation. In this paper, a relatively mature piezoelectric module is used to numerically simulate the giant magnetostrictive transducer, and it is compared with the finite element governing equation of piezoelectric materials. A more mature piezoelectric module is used to simulate the giant magnetostrictive transducer considering the three losses in this paper by comparing the FEM governing equations with piezoelectric materials.

The FEM piezoelectric model can be based on the matrix equation set:(35)$$\begin{array}{l} \left\{S\right\}=\left[S^{E}\right]\left\{T\right\}+\left[d_{piezo}\right]\left\{E\right\}\\ \left\{D\right\}=\left[d_{piezo}\right]\left\{T\right\}+\left[\varepsilon^{T}\right]\left\{E\right\} \end{array}$$

The FEM magnetostrictive model can be based on the matrix equation set:(36)$$\begin{array}{l} \left\{S\right\}=\left[S^{H}\right]\left\{T\right\}+\left[d_{mag}\right]\left\{H\right\}\\ \left\{B\right\}=\left[d_{mag}\right]\left\{T\right\}+\left[\mu^{T}\right]\left\{H\right\} \end{array}$$

Assuming that a sinusoidal voltage is applied to the piezoelectric ceramic electrodes, the FEM governing equation of piezoelectric coupling is [28]
(37)$$\left[\begin{array}{cc} \left[M_{piezo}\right] & \left[0\right]\\ \left[0\right] & \left[0\right] \end{array}\right]\left[\begin{array}{c} \left[\overset{\cdot \cdot }{u}\right]\\ \left[\overset{\cdot \cdot }{V}\right] \end{array}\right]+\left[\begin{array}{cc} \left[C_{piezo}\right] & \left[0\right]\\ \left[0\right] & \left[0\right] \end{array}\right]\left[\begin{array}{c} \left[\overset{\cdot }{u}\right]\\ \left[\overset{\cdot }{V}\right] \end{array}\right]+\left[\begin{array}{cc} \left[K_{piezo}^{E}\right] & \left[K_{{}_{piezo}}^{d}\right]\\ {\left[K_{{}_{piezo}}^{d}\right]}^{T} & \left[K^{\varepsilon }\right] \end{array}\right]\left[\begin{array}{c} \left[u\right]\\ \left[V\right] \end{array}\right]=\left[\begin{array}{c} \left[F\right]\\ \left[Q\right] \end{array}\right]$$
where [*M_piezo_*], [u], [*V*], [*F*], and [*Q*] are the global vectors of the mass matrix, mechanical displacements, electric potential, mechanical force, and electrical charge, respectively. [*C_piezo_*] represents the damping global matrix, and [$K_{piezo}^{E}$], [$K_{piezo}^{d}$], and [$K^{\varepsilon }$] are the global matrices of the elastic, piezoelectric, and dielectric stiffnesses, respectively. The material parameters in Equation (35) can be written as follows: $$K_{piezo}^{d}=\frac{A_{piezo}d_{piezo}}{L_{piezo}S^{E}},\;K_{piezo}^{E}=\frac{A_{piezo}}{L_{piezo}S^{E}},\;K_{piezo}^{\varepsilon }=\frac{\varepsilon^{T}A_{piezo}N_{p}^{2}}{L_{piezo}S^{E}}$$
where *A_piezo_* and *L_piezo_* are the cross-section area and the length of the piezoelectric ceramics, respectively. *N_p_* is the number of piezoelectric ceramics. The superscript T indicates the transpose matrix.

The FEM governing equation of magnetostrictive coupling is
(38)$$\left[\begin{array}{cc} \left[M_{mag}\right] & \left[0\right]\\ \left[0\right] & \left[0\right] \end{array}\right]\left[\begin{array}{c} \left[\overset{\cdot \cdot }{u}\right]\\ \left[\overset{\cdot \cdot }{I}\right] \end{array}\right]+\left[\begin{array}{cc} \left[C_{mag}\right] & \left[0\right]\\ \left[0\right] & \left[0\right] \end{array}\right]\left[\begin{array}{c} \left[\overset{\cdot }{u}\right]\\ \left[\overset{\cdot }{I}\right] \end{array}\right]+\left[\begin{array}{cc} \left[K_{mag}^{H}\right] & \left[K_{mag}^{d}\right]\\ {\left[K_{mag}^{d}\right]}^{T} & \left[K^{\mu }\right] \end{array}\right]\left[\begin{array}{c} \left[u\right]\\ \left[I\right] \end{array}\right]=\left[\begin{array}{c} \left[F\right]\\ \left[\phi \right] \end{array}\right]$$
where [*M_mag_*], [u], [*I*], [*F*], and [ϕ] are the global vectors of the mass matrix, mechanical displacements, current, mechanical force, and magnetic flux, respectively. [*C_mag_*] is the damping global matrix, and [$K_{mag}^{H}$], [$K_{mag}^{d}$], and [$K^{\mu }$] are the global matrices of the elastic, piezomagnetic, and permeability stiffnesses, respectively. Similarly, the material parameters in Equation (36) can be written as follows:$$K_{mag}^{H}=\frac{A_{mag}}{L_{mag}S^{H}},\;K_{mag}^{d}=\frac{A_{mag}d_{mag}}{L_{mag}S^{H}},\;K_{mag}^{\mu }=\frac{\mu^{T}A_{mag}N_{g}^{2}}{L_{mag}S^{H}}.$$
where *A_mag_* and *L_mag_* are the cross-section area and the length of the Terfenol-D rod, respectively. Here, *N_g_* represents the number of coil turns.

By performing a term-by-term comparison of Equations (37) and (38), their mathematical expressions are found to be the same. If we let voltage represent current and current represent voltage, then admittance in the piezoelectric simulation can represent impedance in the magnetostrictive simulation provided that the transduction sections are under the same length and cross-sectional area. Further details include the replacement of the c dielectric constant by the permeability ([$\varepsilon^{T}$] => [*μ^T^*]), the elastic modulus by the elastic modulus ([*S^E^*] => [*S^H^*]), and the $d_{piezo}$ constant by the $d_{mag}$ constant ([*d_piezo_*] *=*> [*d_mag_*]) [7].

Therefore, if losses are considered, the equivalent relationship between the complex parameters of piezoelectric materials [29] and those of giant magnetostrictive materials can be obtained as follows:(39)$$\left\{ \begin{array}{l} S^{E}=S^{E^{\prime }}+jS^{E^{\prime \prime }}\Rightarrow S^{H}=S^{H^{\prime }}+jS^{H^{\prime \prime }}\\ d_{piez}^{}=d_{piez}^{\prime }+d_{piez}^{\prime \prime }\Rightarrow d_{mag}=d_{mag}^{\prime }+jd_{mag}^{\prime \prime }\\ \varepsilon^{T}=\varepsilon^{T^{\prime }}+j\varepsilon^{T^{\prime \prime }}\Rightarrow \mu^{T}=\mu^{T^{\prime }}+j\mu^{T^{\prime \prime }} \end{array}\right.$$

When these material parameters are changed into complex quantities, Equations (36) and (37) remain unchanged, although the matrices [*K^E^_piezo_*], [$K_{piezo}^{d}$], [$K^{\varepsilon }$], [$K_{mag}^{H}$], [$K_{mag}^{d}$], and [$K^{\mu }$] turn to complex number matrices. 

COMSOL Multiphysics 5.5 can be used for the multi-field coupling calculation of transducers, in which the piezoelectric module integrates the losses by setting complex piezoelectric material parameters. We use the parameter comparison method mentioned above to carry out the simulation of the giant magnetostrictive transducer.

Admittedly, this FEM model with complex “piezoelectric material parameters” has its limitations, as the calculation does not include the magnetic circuit of the transducer (the magnetic leakage is ignored), does not consider the coil loss, and so on. 

## 5. Experimental and Simulation Results

To verify the feasibility of using complex parameters to consider the material losses, this paper adopts the widely used FEM software COMSOL Multiphysics 5.5. The main components, such as the head, tail, and Terfenol-D, of the transducer are modeled, ignoring the effects of permanent magnets and disc springs, and transducer screws. In the simulation, meshing is one of the most important aspects to obtain accurate results, especially for wave generation. To accurately resolve sound pressure waves in inner water, this paper specifies the maximum grid cell size as 1/5 of the corresponding minimum wavelength, which is the speed of sound in water (1500 m/s) divided by the maximum frequency used in the frequency sweep. The perfectly matched layer (PML) is meshed using the sweep feature to create a five-layer-structured mesh. Additionally, a layer of “Boundary Layer Mesh” is created within the inner water adjacent to the outer field boundary, with a thickness set to 1/100 of the corresponding minimum wavelength. This boundary layer produces a smooth transition between the inner free tetrahedral mesh and the outer structured prismatic mesh elements, resulting in a higher accuracy of the external field calculation. The finite element model is shown in Figure 9.

The key material parameters (33 directions) are the same as in the PWM equivalent circuit. These material properties are based on the impedance measurements of a Terfenol-D device and extracted by a curve fitting of the measured data [30], and the input material parameters in the simulation are shown in Table 3.

A modal simulation was conducted; the longitudinal vibration mode frequency was 1458 Hz (as shown in Figure 10a), and the flexural mode frequency was 2343 Hz (as shown in Figure 10b). The longitudinal vibration mode was a piston mode, where the radiation head moved forward and backward together in the polarization direction. In the flexural mode, the corners of the radiating head moved in opposite directions to the center. This is the flexural mode of the head mass, which, in general, adversely affects transducer performance [6]. Here, we mainly discussed the simple longitudinal vibration mode to validate the modeling techniques using the complex material quantities. The sound pressure distribution of the transducer in the longitudinal vibration mode is shown in Figure 10c. The current response sent by the transducer in water can be calculated according to the far-field sound pressure value.

### 5.1. Measurement in the Air

Measurements were first made in the air to measure the transducer’s small-signal impedance. An impedance analyzer (Agilent Model E4990A) was used to measure the impedance of the transducer as a function of frequency.

It can be seen in Figure 11a from the impedance spectrum that the improved PWM and FEM models considering the material losses (without eddy loss) roughly approach the measured impedance curve. There are still a few discrepancies in the phase curves across the wide frequency range. The PWM and FEM simulation results after further consideration of the eddy current loss (Figure 11b) produce more accurate results compared with the measured results. The eddy current loss is frequency-dependent and increases with the frequency, so the permeability should vary with the frequency. The eddy current factor was introduced to make the relative permeability in the model be frequency-dependent, so the frequency dependence of the eddy current loss can be considered. Therefore, the phase curves of both simulation models considering the frequency dependence of the eddy current losses are closer to the experimental measurements. Both results are in good agreement with the impedance and phase curves.

### 5.2. Underwater Measurement

To yield better underwater measurement results, the experiment was carried out in a lake with a depth of 100 m and a minimum diameter of 2 km. The transducer was only deployed at a 10 m depth to ensure that the pre-stress stayed around 20 MPa in the lake, along with a standard hydrophone (B&K 8104) positioned 3.2 m away from the center of the radiating head of the transducer. A simplified schematic diagram of the transducer’s underwater test is shown in Figure 12. The experimental settings are shown in Figure 13 and Figure 14. The output voltages and currents of the power amplifier, the phase difference between them, and the hydrophone output voltage were measured and saved. The large-signal impedance and TCR were calculated.

The simulated results of the improved PWM and FEM model and the impedance curves measured in the experiments are shown in Figure 15. It should be noted that the coil loss, measured with an impedance analyzer without the insertion of the Terfenol-D rods, was also taken into account by adding it to both simulated results. The frequency range was 800–1300 Hz. It can be seen from the impedance curve that, after considering the eddy current loss, the impedance amplitudes calculated using the PWM and FEM models are very close to the experimental measurement values. The impedance curve produced by the FEM simulation agrees well with the measurement.

TCR was one of the most important design criteria for transducer design because the operating frequencies and the mechanical quality factors were determined with respect to the TCR results. When a 1 A alternating current was applied to the PWM circuit shown in Figure 8, the current in the electrical domain became analogous to the velocity of the active surface in the mechanical domain. After calculating the “current” in the PWM circuit, the pressure in the far-field distance can be obtained by using the following Equation (7):(40)$$|P_{r}|=\frac{\rho ckuR^{2}}{2r}$$
where ρ is the density of the medium, *c* is the speed of sound in the medium, *u* is the speed of the active surface, and *r* is the distance of the far field. After calculating the far-field pressure, the TCR is calculated as follows:(41)$$TCR=20\log(\frac{P_{rms}}{P_{ref}})$$
where *P_rms_* is the root mean square pressure obtained in the far field, and *P_ref_* is the reference pressure, which is 1 μPa for water.

The simulation and measurement results of TCR are shown in Figure 16. It can be seen that the maximum TCR results of the improved PWM and FEM models established in this paper are close to the measured results, and the prediction accuracy is shown in Table 4. The prediction accuracy of the FEM model (with the eddy current loss) for the maximum TCR is 0.32%, and the prediction accuracy of the resonance point is 1%. This high prediction accuracy is very rare for the PWM of the transducer, which validates the proposed method of using the complex material parameters. Figure 16 also shows that the simulation results of the PWM and FEM models have a slight discrepancy. This may be attributed to the fact that the PWM model in this paper is a one-dimensional model, while FEM is a three-dimensional model. The discrepancies over the frequency range, except for at the resonance frequency, might result from the fluid simulation. FEM have better accuracies on the acoustic field. In the FEM simulation, a full sound absorption (no reflection) condition was applied at the outer boundary of the water medium. For the calculation of the TCR, a receiver node was set to read complex sound pressures at a distance approximately in a far field from the fluid–structure interaction layer.

## 6. Conclusions

This paper proposes improved equivalent PWM and FEM models considering the elastic loss, piezomagnetic loss, and magnetic loss of Terfenol-D. On this basis, the frequency-dependent eddy current losses are also considered and incorporated into the two models. To verify the accuracy of the established PWM and FEM models considering the three losses, a high-power Tonpilz Terfenol-D transducer with a resonance frequency of around 1 kHz and a maximum transmitting current response (TCR) of 187 dB/1A/μPa is manufactured and tested. A comparison of the models with/without considering the losses with the experimental results shows the necessity and importance of considering the material losses. The PWM and FEM simulation results considering the loss are in good agreement with the experimental results, especially the model considering the variation of eddy current loss with frequency has a high agreement with the experimental test results (the relative error between the TCR of the PWM model at resonance and the experimental results is only 0.37%, and FEM is only 0.32%); this indicates that the PWM and FEM models established in this paper considering the losses are suitable for giant magnetostrictive transducers. Under a small signal, the eddy current loss is small, so the models without the eddy current losses could still fit with the experiment results with a certain accuracy. However, under 1 A current excitation, the eddy current losses are more significant, so the simulation must consider the frequency-dependent eddy current losses to closely fit the experimental curve. Our proposed method may shed light on the more predictable design of high-power giant magnetostrictive transducers in the future.

## Figures and Tables

**Figure 1 sensors-22-03781-f001:**
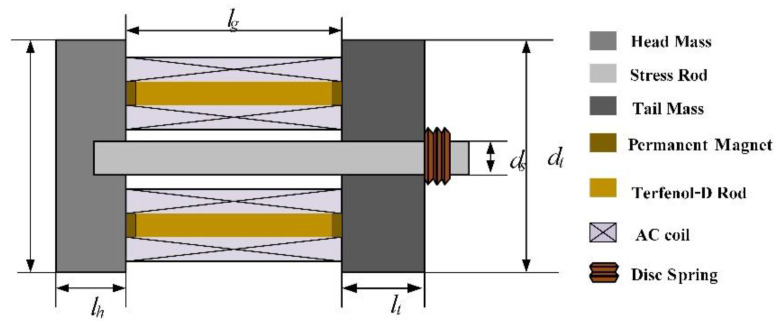
Cross-section view of the Tonpilz Terfenol-D.

**Figure 2 sensors-22-03781-f002:**
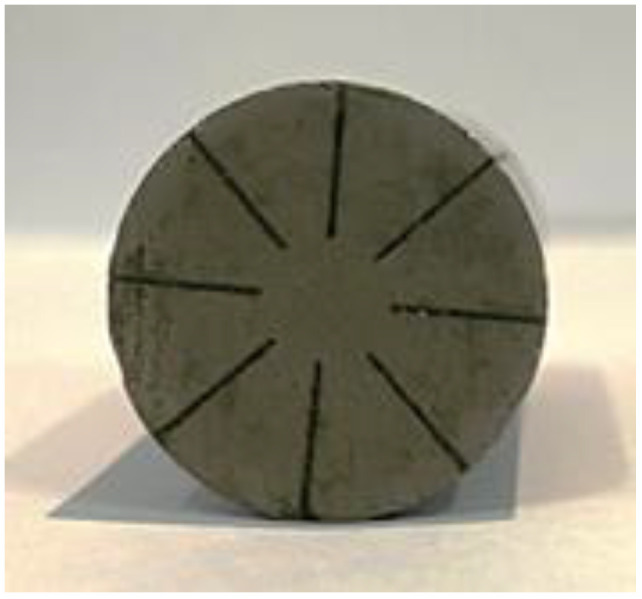
Photo of a radially slotted Terfenol-D rod.

**Figure 3 sensors-22-03781-f003:**
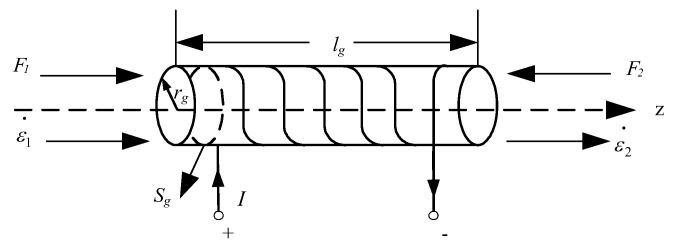
Configuration of a *k*_33_ sample.

**Figure 4 sensors-22-03781-f004:**
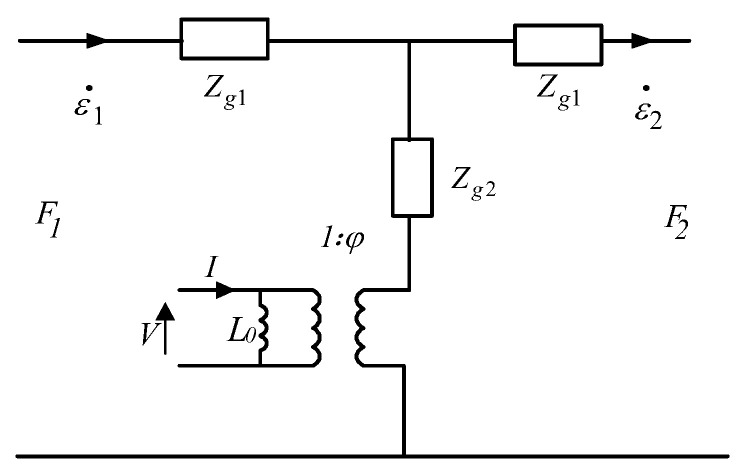
PWM equivalent circuit without material losses.

**Figure 5 sensors-22-03781-f005:**
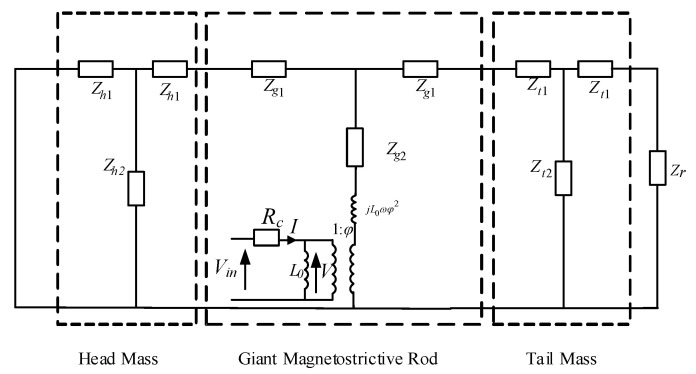
The electromechanical equivalent circuit of the Tonpilz Terfenol-D transducer without material losses.

**Figure 6 sensors-22-03781-f006:**
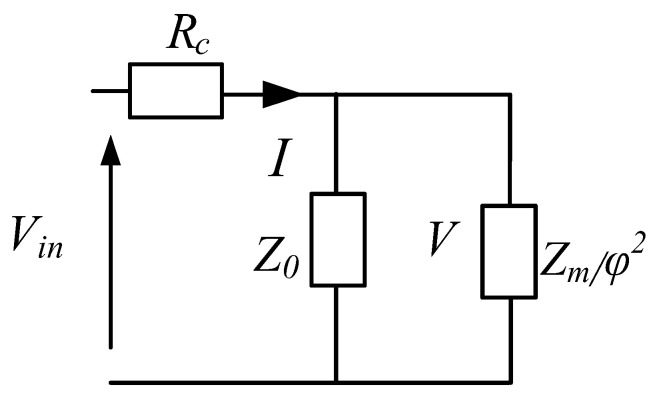
Simplified equivalent circuit of the Tonpilz Terfenol-D.

**Figure 7 sensors-22-03781-f007:**
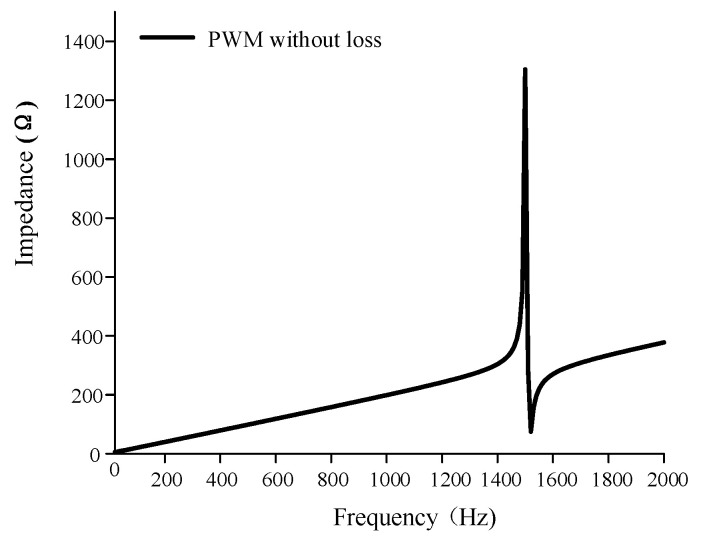
Simulated impedance of the Tonpilz Terfenol-D transducer without material losses.

**Figure 8 sensors-22-03781-f008:**
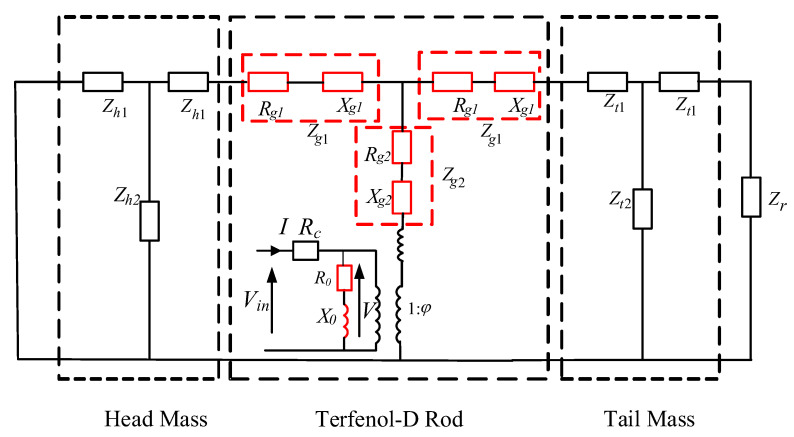
The equivalent PWM circuit of the Tonpilz Terfenol-D transducer considering material losses.

**Figure 9 sensors-22-03781-f009:**
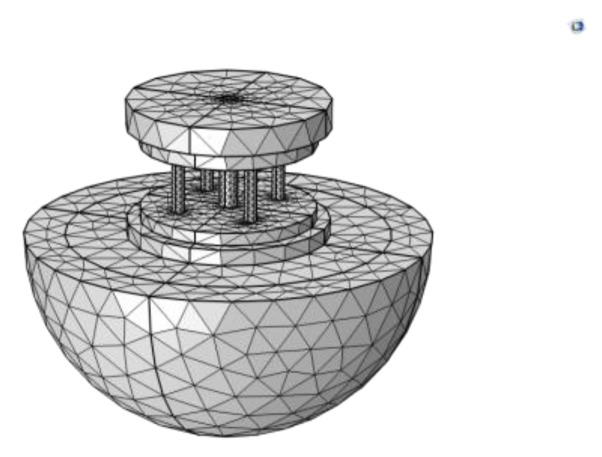
Schematic diagram of transducer modeling.

**Figure 10 sensors-22-03781-f010:**
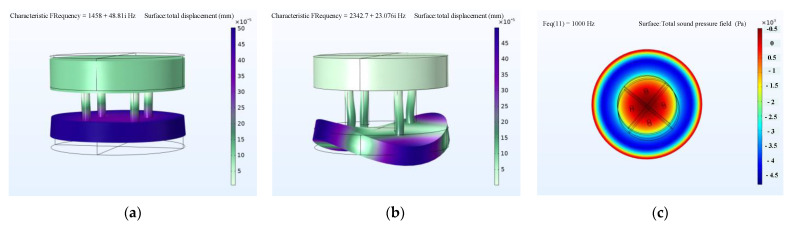
Modal and harmonic response simulation of Tonpilz Terfenol-D transducer: (**a**) longitudinal vibration mode of the Tonpilz Terfenol-D transducer by FEM; (**b**) flexural vibration mode of the Tonpilz Terfenol-D transducer by FEM; (**c**) results of sound field of the Tonpilz Terfenol-D transducer by FEM.

**Figure 11 sensors-22-03781-f011:**
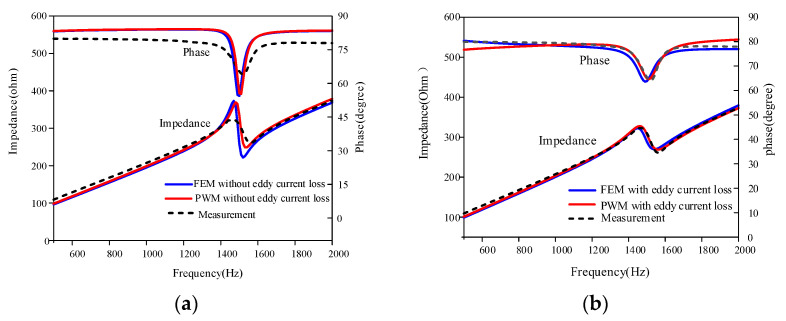
Impedance spectrum in air: (**a**) impedance spectrum of the simulation and experiment without considering eddy loss; (**b**) impedance spectrum of the simulation and experiment considering eddy loss.

**Figure 12 sensors-22-03781-f012:**
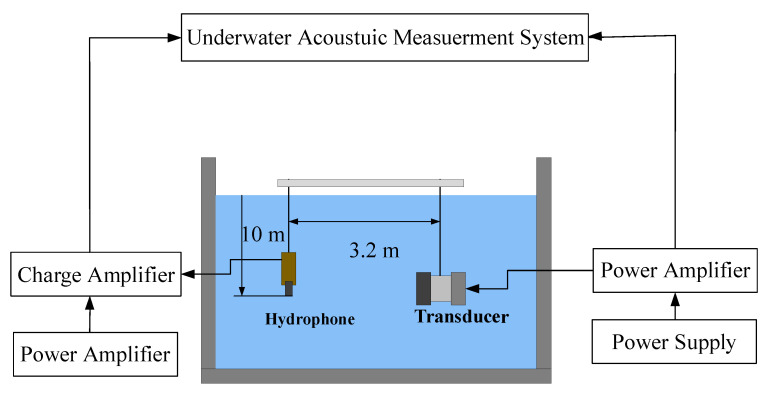
Schematic of underwater measurement system for transducer.

**Figure 13 sensors-22-03781-f013:**
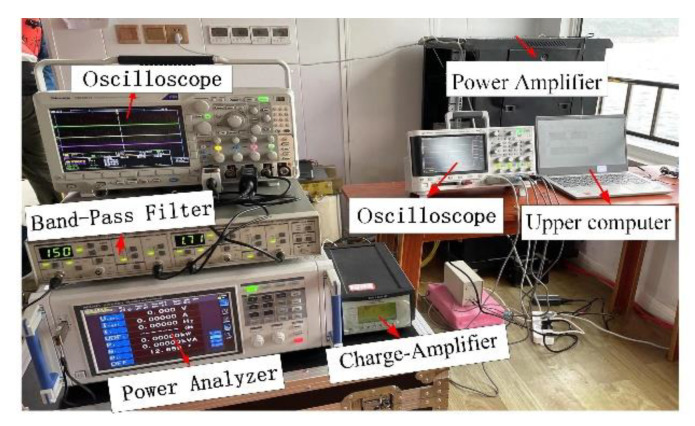
Underwater acoustic measurement system.

**Figure 14 sensors-22-03781-f014:**
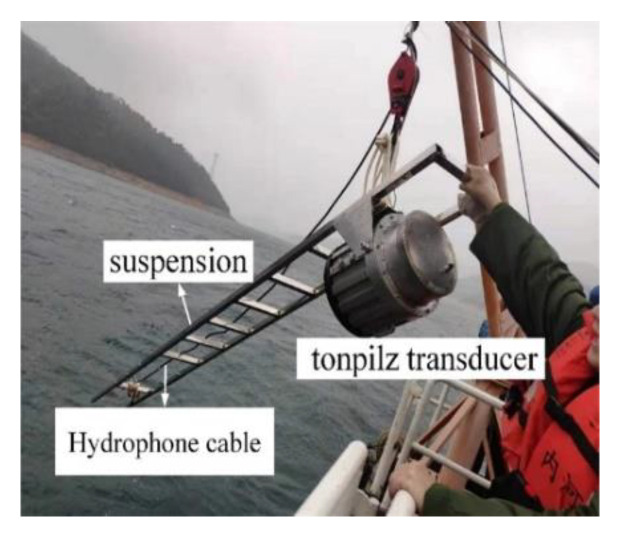
The transducer in water.

**Figure 15 sensors-22-03781-f015:**
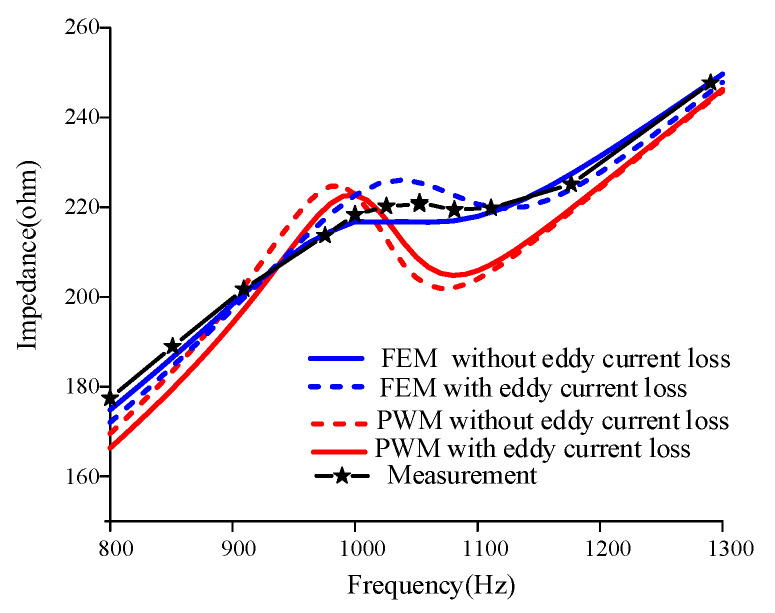
Comparison of the impedance for PWM, FEM, and measurement.

**Figure 16 sensors-22-03781-f016:**
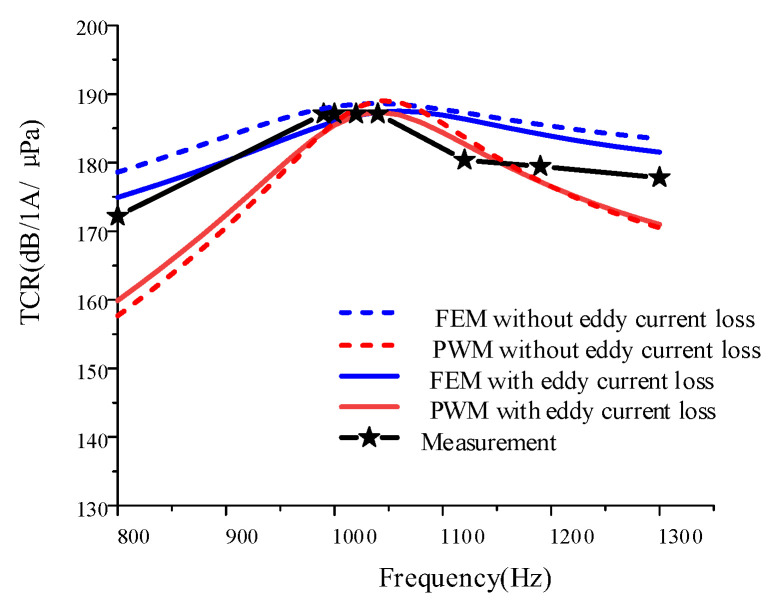
Comparison of the TCR for PWM, FEM, and measurement.

**Table 1 sensors-22-03781-t001:** Structural parameters of the transducer.

Structure	Material	Density (kg/m^3^)	Radius (mm)	Thickness (mm)
Head mass	Aluminum	2700	160	45
Terfenol-D Rod	Terfenol-D	9250	10	100
Tail mass	Stainless steel	7900	160	72

**Table 2 sensors-22-03781-t002:** PWM equivalent circuit parameters.

	Terfenol-D	Tail Mass	Head Mass
**Distributed impedance**	$Z_{g1}=j\rho_{g}c_{g}S_{g}\tan(\frac{k_{g}l_{g}}{2})$	$Z_{t1}=j\rho_{t}c_{t}S_{t}\tan(\frac{k_{t}l_{t}}{2})$	$Z_{h1}=j\rho_{h}c_{h}S_{h}\tan(\frac{k_{h}l_{h}}{2})$
**Distributed impedance**	$Z_{g2}=\rho_{g}c_{g}S_{g}/j\sin(k_{g}l_{g})$	$Z_{t2}=\rho_{t}c_{t}S_{t}/j\sin(k_{t}l_{t})$	$Z_{h2}=\rho_{h}c_{h}S_{h}/j\sin(k_{h}l_{h})$
**Wave number**	$k_{g}=\frac{\omega }{c_{g}}$	$k_{t}=\frac{\omega }{c_{t}}$	$k_{h}=\frac{\omega }{c_{h}}$
**Wave speed (m/s)**	$c_{g}=\frac{1}{\sqrt{\rho_{g}S_{33}^{H}}}$	$c_{t}=4942$	$c_{h}=5128$

**Table 3 sensors-22-03781-t003:** Material properties of the Tonpilz transducer.

Material	Material Properties
**Head mass (Aluminum)**	**Density:** 2700 kg/m^3^
	**Young’s modulus:** 71 GPa
	**Poisson’s ratio:** 0.33
**Tail mass (Stainless steel)**	**Density:** 7900 kg/m^3^
	**Young’s modulus:** 193 GPa
	**Poisson’s ratio:** 0.28
**Terfenol-D**	**Density:** 9250 kg/m^3^
	$\mathrm{Flexibility}\;\mathrm{coefficient}:\;S_{33}^{B^{\prime }}=1.65\times 10^{-11}$; $\;S_{33}^{B^{\prime \prime }}=-5.48\times 10^{-13}$
	$\mathrm{Piezomagnetic}\;\mathrm{coefficient}:\;d_{33}^{\prime }=7.4\times 10^{-10}$; $\;d_{33}^{\prime \prime }=-2.2\times 10^{-12}$
	$\mathrm{Relative}\;\mathrm{permeability}:\;\mu_{33}^{T^{\prime }}=4.58$; $\;\mu_{33}^{T^{\prime \prime }}=-0.66$

**Table 4 sensors-22-03781-t004:** The parameters calculated by modeling methods and their relative deviations (%) from the measurement results.

	PWM (without Eddy Current Losses)	PWM (with Eddy Current Losses)	FEM (without Eddy Current Losses)	FEM (with Eddy Current Losses)	Measurement
***f_r_* (Hz)**	1045 (4.5%)	1020 (2%)	1015 (1.5%)	1010 (1%)	1000
**Max TCR (dB/1A/μPa)**	190 (1.55%)	187.8 (0.37%)	189.8 (1.4%)	187.7 (0.32%)	187

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
