# Peer review of "Modeling of High-Power Tonpilz Terfenol-D Transducer Using Complex Material Parameters"

_sensors, 2022, doi:10.3390/s22103781_

Round 1
Reviewer 1 Report
Paper can be accepted after the following corrections:
- Acronyms: FEM, PWR and TCR should be explained.
- Details about the FEM modelling should be provided. Please provide information concerning modelling software, meshes and convergence criteria.
- Conclusions should be developed to clearly present the most important results in the paper.
Reviewer 2 Report
This manuscript reports the loss effects of the Terfenol-D material, an active part of transducers, with important implications for their characteristics. The elastic, piezomagnetic, magnetic losses and the frequency-dependent eddy current reported for the high-power low-frequency transducers become appreciable. The theoretical approach improving by considering these complex parameters is efficient to refine the popular plane wave method (PWM) circuit model and finite element method (FEM) model for a predictable design. The studies have been relatively well performed and are well described in this paper. From my point of view, it is a pleasure to recommend the manuscript for publication in Sensors.
Reviewer 3 Report
The authors model high-power Tonpilz transducers based on Terfenol-D. First, they establish an equivalent circuit which takes into account losses. Second, using an analogy between piezoelectricity and piezomagnetism, they use a commercial software (COMSOL) to simulate the magnetostrictive transducer. Finally, the results of modeling are compared with experiments.
The paper is in general clearly written and comprehensive, it includes analytical modeling, numerical simulations and expereiments. I have just a few minor comments, which can help to improve the manuscript.
1) Introduction --> Please explain your abbrevations (PWM, FEM, TCR etc.) at first time when they appear in the text.
2) Line 22 --> Do you mean thermal expansion coefficient? Please specify it.
3) Line 23 --> "High magnet-mechanical coupling coefficient" --> Please define this coefficient anf give a number (what is "high"?); "high energy density" --> Please give a number (what is "high"?)
4) Figure 1 --> Letters are too smal. The Figure is difficult to read. Please redraw.
5) 3. Loss-integration into PWM equivalent circuit --> Please include a statement if this consideration is original or it has been already reported in the work of others.
6) Please explain the physical meaning of the imaginary part of the inductance. You seem to use the engineering convention for harmonic time dependence (e^[j*omega*t]). Please double check the sign before the imaginary part of the inductance in Eq. (18) (and the signs in Eqs. (17)). They may depend on the selected convention for the time dependence. The resistances representing losses must be always positive. Please write down explicitly your convention for the time dependence.
7) Figure 12 --> What is an "upper computer"? Please reconsider your choice of notation.
8) Line 337 --> It should be "design".
9) 4. Loss-integration into FEM --> Please include references to the previous works pointing onto the analogy between piezoelectric and piezomagnetic phenomena.
Round 2
Reviewer 1 Report
The paper was corrected and can be accepted in the present form.